# Increased C-X-C Motif Chemokine Ligand 12 Levels in Cerebrospinal Fluid as a Candidate Biomarker in Sporadic Amyotrophic Lateral Sclerosis

**DOI:** 10.3390/ijms21228680

**Published:** 2020-11-17

**Authors:** Pol Andrés-Benito, Mònica Povedano, Raúl Domínguez, Carla Marco, Maria J. Colomina, Óscar López-Pérez, Isabel Santana, Inês Baldeiras, Sergio Martínez-Yelámos, Inga Zerr, Franc Llorens, Joaquín Fernández-Irigoyen, Enrique Santamaría, Isidro Ferrer

**Affiliations:** 1Department of Pathology and Experimental Therapeutics, University of Barcelona, Feixa Llarga s/n, 08907 L’Hospitalet de Llobregat, Barcelona, Spain; franc.llorens@gmail.com; 2Biomedical Network Research Center on Neurodegenerative Diseases (CIBERNED), Institute Carlos III, Feixa Llarga s/n, 08907 L’Hospitalet de Llobregat, Barcelona, Spain; oscarlzpz@gmail.com; 3Bellvitge Biomedical Research Institute (IDIBELL), 08907 L’Hospitalet de Llobregat, Barcelona, Spain; 4Institute of Neurosciences, University of Barcelona, 08907 L’Hospitalet de Llobregat, Barcelona, Spain; 5International Initiative for Treatment and Research Initiative to Cure ALS (TRICALS), Bellvitge University Hospital, 08907 Hospitalet de Llobregat, Spain; mpovedano@bellvitgehospital.cat (M.P.); rdominguez@bellvitgehospital.cat (R.D.); carlamarcocazcarra@gmail.com (C.M.); 6Functional Unit of Amyotrophic Lateral Sclerosis (UFELA), Service of Neurology, Bellvitge University Hospital, 08907 L’Hospitalet de Llobregat, Barcelona, Spain; 7Anesthesia and Critical Care Department, Bellvitge University Hospital-University of Barcelona, 08907 L’Hospitalet de Llobregat, Barcelona, Spain; Mjcolomina@gmail.com; 8Neurology Department, CHUC—Centro Hospitalar e Universitário de Coimbra, CNC—Center for Neuroscience and Cell Biology; and Faculty of Medicine, University of Coimbra, 3000-456 Coimbra, Portugal; isabeljsantana@gmail.com (I.S.); ines.baldeiras@sapo.pt (I.B.); 9Multiple Sclerosis Unit, Service of Neurology, Bellvitge University Hospital, 08907 L’Hospitalet de Llobregat, Barcelona, Spain; smartinezy@bellvitgehospital.cat; 10Department of Neurology, University Medical Center Göttingen, 37075 Göttingen, Germany; ingazerr@med.uni-goettingen.de; 11German Center for Neurodegenerative Diseases (DZNE), 37075 Göttingen, Germany; 12IDISNA, Navarra Institute for Health Research, 31008 Pamplona, Spain; joaquin.fernandez.irigoyen@navarra.es (J.F.-I.); enrique.santamaria.martinez@navarra.es (E.S.); 13Clinical Neuroproteomics Unit, Proteomics Platform, Proteored-ISCIII, Navarrabiomed, Complejo Hospitalario de Navarra (CHN), Universidad Pública de Navarra (UPNA), 31008 Pamplona, Spain; 14Neuropathology, Pathologic Anatomy Service, Bellvitge University Hospital, 08907 L’Hospitalet de Llobregat, Barcelona, Spain

**Keywords:** amyotrophic lateral sclerosis, cerebrospinal fluid, proteomics, biomarkers, CXCL12, CXCR4, CXCR7, AAAS, S1006A

## Abstract

Sporadic amyotrophic lateral sclerosis (sALS) is a fatal progressive neurodegenerative disease affecting upper and lower motor neurons. Biomarkers are useful to facilitate the diagnosis and/or prognosis of patients and to reveal possible mechanistic clues about the disease. This study aimed to identify and validate selected putative biomarkers in the cerebrospinal fluid (CSF) of sALS patients at early disease stages compared with age-matched controls and with other neurodegenerative diseases including Alzheimer disease (AD), spinal muscular atrophy type III (SMA), frontotemporal dementia behavioral variant (FTD), and multiple sclerosis (MS). SWATH acquisition on liquid chromatography-tandem mass spectrometry (LC–MS/MS) for protein quantitation, and ELISA for validation, were used in CSF samples of sALS cases at early stages of the disease. Analysis of mRNA and protein expression was carried out in the anterior horn of the lumbar spinal cord in post-mortem tissue of sALS cases (terminal stage) and controls using RTq-PCR, and Western blotting, and immunohistochemistry, respectively. SWATH acquisition on liquid chromatography-tandem mass spectrometry (LC–MS/MS) revealed 51 differentially expressed proteins in the CSF in sALS. Receiver operating characteristic (ROC) curves showed CXCL12 to be the most valuable candidate biomarker. We validated the values of CXCL12 in CSF with ELISA in two different cohorts. Besides sALS, increased CXCL12 levels were found in MS but were not altered in AD, SMA, and FTD. Therefore, increased CXCL12 levels in the CSF can be useful in the diagnoses of MS and sALS in the context of the clinical settings. CXCL12 immunoreactivity was localized in motor neurons in control and sALS, and in a few glial cells in sALS at the terminal stage; CXCR4 was in a subset of oligodendroglial-like cells and axonal ballooning of motor neurons in sALS; and CXCR7 in motor neurons in control and sALS, and reactive astrocytes in the pyramidal tracts in terminal sALS. CXCL12/CXCR4/CXCR7 axis in the spinal cord probably plays a complex role in inflammation, oligodendroglial and astrocyte signaling, and neuronal and axonal preservation in sALS.

## 1. Introduction

Amyotrophic lateral sclerosis (ALS) is a devastating neurodegenerative disease characterized by progressive loss of upper and lower motor neurons [1]. ALS may rarely be categorized as sporadic ALS (sALS) due to point mutations, and familial ALS which is linked to mutations in a large variety of apparently unrelated genes.

Increased levels of neurofilaments (NF) in the cerebrospinal fluid (CSF) are considered useful biomarkers in ALS [2,3]. NF heavy chain levels in CSF are negatively correlated with disease duration and ALS functional rating scale-revised (ALS-FRS-R); NF light chain levels in CSF are negatively correlated with disease duration. Increased levels of NF-L in the CSF are not specific to the disease, but they correlate with disease progression [3,4,5]. Several cytokines and mediators of the inflammatory response are also increased in the CSF in ALS [6,7]. However, the protein list is variable from one study to another which makes it difficult to use any of them as a unique CSF biomarker of inflammation in ALS. This is due in part to the different methods employed in different laboratories and also to the respiratory status of the patient [8]. Finally, increased levels of YKL40 in the CSF correlate with disease progression [4,9,10,11]. Since YKL40 is expressed in inflammatory astrocytes [4,12], YKL40 in the CSF indicates the final balance of YKL40 metabolism in inflammatory astrocytes, and its delivery to the CSF in ALS.

CSF proteomics is a very useful tool in discovering new putative biomarkers in ALS. Specifically, we have used SWATH-mass spectrometry (where SWATH is sequential window acquisition of all theoretical mass spectra), intending to obtain high-throughput, accurate quantification, and reproducible measurements across CSF proteomes derived from the CSF in the first cohort of sALS cases at the beginning of clinical symptoms (within the first year after the appearance of the first symptoms). To our knowledge, this is the first study where SWATH-based proteomic analysis is applied to explore novel CSF biomarkers related to sALS. To complement the pipeline, an orthogonal approach such as ELISA was used to validate the expression of CXCL12 as a putative marker. Correlation analysis between protein expression levels and clinical parameters including age at onset, gender, first signs, and progression of the disease following well-known clinical scales were performed. Next, we studied the expression of mRNA and protein levels in the post-mortem anterior horn of the spinal cord in another series of sALS cases at terminal stages of the disease and localized the presence of the protein and its receptors with immunohistochemistry. Then we tested CXCL12 CSF levels in a validation (second) cohort of sALS cases. Finally, a cross-disease study was performed to monitor CXCL12 levels in sALS, multiple sclerosis (MS), Alzheimer’s disease (AD), frontotemporal dementia behavioral variant (FTD), and type III spinal muscular atrophy (SMA).

## 2. Results

### 2.1. Proteomics Results

We conducted a SWATH-based proteomic analysis to explore novel CSF biomarkers related to ALS, and to determine whether proteomic changes can provide new insight into ALS pathophysiology. For that, CSF proteomics was performed on 15 sALS and 15 controls from the first cohort selected at random revealing 51 differentially expressed proteins applying a cut-off *p*-value < 0.05, and absolute fold changes of <0.77 (down-regulation) or >1.3 (up-regulation) in linear scale (Figure 1). A heat-map represents the intensity of changes in the differentially expressed proteins (Figure 1; Table 1). The total number of proteins quantified by SWATH MS in all CSF samples was 1468. The complete list of quantified proteins can be obtained upon request.

### 2.2. Identification of Putative Biomarker Candidates and Quantification Using ELISA

Based on the statistical significance in the exploratory proteomics study and the putative biological relevance, altered protein CSF levels were assessed in the first cohort of 43 sALS cases and 36 controls using commercial ELISA kits. We selected CXCL12/SDF1α (FC = 1.97, *p* = 0.0001) for its role in the immune response; aladin (AAAS) (FC = 2.19, *p* = 0.024) for its implication as an active component of nuclear envelope and nuclear transport, calcyclin (S100A6) (FC = 1.67, *p* = 0.028) for its implication in calcium signaling and inflammation, PAAF1 (FC = 0.425, *p* = 0.034) as a regulatory element of the proteasome; and syntaxin-12 (FC = 2.01, *p* = 0.037) as a component of neurotransmission and endosomal fusion.

Significantly higher CXCL12 protein levels were detected in sALS (593.00 ± 33.08 pg/mL) compared with controls (305.00 ± 21.5 pg/mL) (*p* = 0.000) (Figure 2A). To calculate the clinical accuracy of CXCL12 in discriminating between sALS and the control group, we estimated the AUC value of 0.896 ± 0.037, 95% CI: 0.821–0.968 (*p* = 0.000). Considering the optimal cut-off at 469.03 pg/mL, defined by the Youden index (0.705), an overall sensitivity of 76.92% and specificity of 93.55% were predicted.

In contrast, AAAS levels were reduced in sALS (2123.00 ± 55.29 pg/mL) when compared with controls (2836.00 ± 82.51 pg/mL) (*p* = 0.000) (Figure 2B). Statistical analysis revealed an AUC value of 0.895 ± 0.0342, 95% CI: 0.827–0.962; considering the optimal cut-off at 2559 pg/mL, defined by the Youden index (0.654), an overall sensitivity of 92.68% and specificity of 72.73% were predicted.

Significantly decreased levels of S100A6 were found in sALSs (715.30 ± 34.88 pg/mL) when compared with controls (1021.00 ± 88.73 pg/mL) (*p* = 0.005) (Figure 2C). Statistical analysis revealed AUC of 0.687 ± 0.0659, 95% CI: 0.5578–0.8161. Considering the optimal cut-off at 1002.00 pg/mL, defined by the Youden index (0.361), an overall sensitivity of 90.91% and specificity of 45.16% can be predicted.

PAAF1 levels were significantly increased in sALS (7.07 ± 0.32 pg/mL) when compared with controls (4.80 ± 0.18 pg/mL) (*p* = 0.000) (Figure 2D). Statistical analysis revealed an AUC value of 0.895 ± 0.033, 95% CI: 0.831–0.961. Considering the optimal cut-off at 5.68 pg/mL, defined by the Youden index (0.612), an overall sensitivity of 84.09% and specificity of 77.14% were predicted.

SNTX12 levels were significantly increased in sALS (17.07 ± 1.12 pg/mL) when compared with controls (9.18 ± 0.96 pg/mL) (*p* = 0.000) (Figure 2E). Statistical analysis revealed an AUC value of 0.859 ± 0.052, 95% CI: 0.756–0.96. Considering the optimal cut-off at 14.52 pg/mL, defined by the Youden index (0.635), an overall sensitivity of 77.81% and specificity of 85.71% were predicted.

To establish the best candidate for ALS biomarker, we compared the AUC values of candidates; the results showed that the most likely was CXCL12 the AUC value of which showed significant differences or tendencies when compared with AAAS (*p* = 0.08), S100A6 (*p* = 0.002), PAAF1 (*p* = 0.08) and STX12 (*p* = 0.08) AUC values (Figure 2F). Additionally, based on the high sensitivity of AAAS as a biomarker, we also studied the diagnostic value of CXCL12/AAAS ratio to discriminate between sALS and the control group. Data indicated significant differences when sALS cases were compared (0.24 ± 0.01 AU) with control cases (0.13 ± 0.018 AU) (*p* = 0.000), but the AUC value (0.855 ± 0.47, 95% CI: 0.762–0.947) did not improve the clinical accuracy in discriminating sALS and control cases.

### 2.3. Associations of CSF Biomarker Levels, and Clinical and Biochemical Parameters

CSF levels of studied biomarkers were compared with several clinical variables, including age, sex, type of clinical symptoms at onset, ALS-FSR-R score, diagnosis delay, and disease progression. Associations between biomarker data and these parameters did not reveal significant correlations, except for age which correlated with S100A6 levels in control (Pearson’s r = 0.344; *p* = 0.05) (Figure 3A) and sALS cases (Pearson’s r = 0.575; *p* = 0.000) (Figure 3B). Additionally, biochemical associations with neurofilaments-L (NF-L) and chitinase 3 like 1 (YKL40) [4], revealed that CXCL12 levels did not correlate with control cases (Figure 3C), but correlated positively with YKL40 levels in sALS (Pearson’s r = 0.426; *p* = 0.003) (Figure 3D), but not with NF-L protein levels in controls (Pearson’s r = 0.14; *p* = 0.88) and sALS (Pearson’s r = −0.02; *p* = 0.88) (Figure 3E,F).

### 2.4. Candidate Biomarker mRNA Expression and Protein Levels in the Spinal Cord Tissue of sALS and Controls

We included controls (*n* = 17, mean age ± SD, 64.9 ± 10.6 years) and sALS cases (*n* = 22, mean age ± SD: 62.4 ± 10.8). mRNA levels of CXCL12 (*p* = 0.002) were significantly increased in sALS when compared with controls. A significant increase in CXCR4 mRNA expression was found in sALS when compared with controls (*p* = 0.000) but expression levels of CXCR7 mRNA were not significantly altered (*p* = 0.20). Astrocytic and microglial genes were evaluated, including GFAP, CHI3L1, AIF1, and CD68 transcripts. Increased mRNA levels of CHI3L1 (*p* = 0.000), AIF1 (*p* = 0.000), and CD68 (*p* = 0.000), but not of GFAP (*p* = 0.77), were noted in sALS cases when compared with controls (Figure 4A).

Gel electrophoresis and Western blotting of the anterior horn of the lumbar spinal cord showed a trend of increased CXCL12 protein expression in spinal cord tissue in sALs, but the difference from control was not statistically significant when revealed (*p* = 0.22) (Figure 4B).

Immunohistochemistry revealed that CXCL12 localized in the cytoplasm of motor neurons in control and ALS. Additionally, CXCL12 immunoreactivity was present in a few glial cells in the anterior horn in sALS. CXCR4 was localized in axons in the control spinal cord and axonal swellings of motor neurons, degenerating motor neurons, and in a subset of oligodendroglia-like cells in sALS. CXCR7 immunoreactivity was found in the cytoplasm of motor neurons in control and sALS, and in reactive astrocytes in the pyramidal tracts in sALS (Figure 5).

### 2.5. Increased CSF CXCL12 Protein Levels Distinguish sALS (and MS) from Other Neurodegenerative Diseases

Validation of CXCL12 protein levels in the CSF in sALS was assessed in the validation (second) cohort of 65 sALS cases and 44 controls. Increased CXCL12 protein levels in the CSF were found in sALS cases (n = 65; 662.6 ± 24.85 pg/mL) when compared with controls (n = 44; 307.4 ± 22.91 pg/mL) (*p* = 0.000) (Figure 6A). The accuracy of CXCL12 to discriminate sALS and controls revealed ACU 0.922 ± 0.0247, 95% CI: 0.8738 to 0.9708. Considering the optimal cut-off at 412.15 pg/mL, defined by the Youden index (0.675), the overall sensitivity of 90.00% and specificity of 77.50% were predicted. The pair-wise comparison of Receiver operating characteristic (ROC) curves did not indicate differences between the first cohort and the second (*p* = 0.2162). Thus, CXCL12 protein levels were significantly increased in sALS cases (n = 108; 632.8 ± 19.43 pg/mL) when compared with controls (n = 81; 316.50 ± 16.25 pg/mL) (*p* = 0.000) (Figure 6B). To calculate the accuracy of CXCL12 in distinguishing sALS and controls, the AUC value was assessed: AUC: 0.913 ± 0.0203, 95% CI: 0.86–0.952, *p* = 0.000. Considering the optimal cut-off at 410.00 pg/mL, defined by Youden index (0.641), the overall sensitivity of 89.80% and specificity of 74.32% were predicted. Differences were not seen between the first and the second cohort despite gender differences with the predominance of females in the first cohort.

Finally, to learn about disease specificity of CXCL12 in sALS, the CSF of other neurodegenerative diseases, including SMA (n = 13), AD (n = 19), FTD (n = 39), and MS (n = 30) was analyzed. Significantly higher CXCL12 levels were found in MS patients (700.08 ± 55.68 pg/mL) (*p* = 0.000) but not in AD (244.17 ± 19.43 pg/mL), FTD (269.16 ± 10.48 pg/mL), or SMA-III (310.70 ± 43.70 pg/mL) when compared with controls (316.50 ± 16.25 pg/mL) (Figure 6B). AUC values for CXCL12 levels did not discriminate between SMA (AUC: 0.520 ± 0.098; 95% CI: 0.41 to 0.63, *p* = 0.84), AD (AUC: 0.654 ± 0.085; 95% CI: 0.548 to 0.75, *p* = 0.06), FTD (AUC: 0.572 ± 0.0532; 95% CI: 0.475 to 0.665, *p* = 0.17), and control cases, in contrast with MS (AUC: 0.858 ± 0.057; 95% CI: 0.775 to 0.919, *p* = 0.000) (Figure 6C).

## 3. Discussion

Proteomic analysis of the CSF has identified 51 proteins differentially expressed in the CSF of sALS when compared with age-matched controls. Based on their statistical and biological significance, we selected five proteins and validated their changes in the CSF of the first cohort of sALS. C-X-C motif chemokine ligand 12 (CXCL12/SDF1α), syntaxin-12 (STX12), and proteasomal ATPase associated factor 1 (PAAF1) protein levels are significantly increased, whereas aladin WD repeat nucleoporin (AAAS) and S100 calcium-binding protein A6 (S100A6) are significantly reduced in CSF of sALS. ROC curves have shown CXCL12 the most interesting biomarker candidate, the AUC values of which showed significant differences or tendencies when compared with AAAS, S100A6, PAAF1, and STX12. For this reason, CXCL12 was chosen for additional study. The opportunity to study other proteins identified in the proteomics study will be assessed in the future. For example, HBB is significantly reduced in ALS as measured by seven peptides. This is likely due to high hemoglobin levels in control samples than sALS. It would be interesting to measure the correlation of other proteins to HBB, which may indicate if any were detected due to blood contamination. Blood-brain barrier leakage is a possible explanation as well. However, reduced CSF hemoglobin might parallel reduced hemoglobin in neurons in sALS as well.

Stromal cell-derived factor 1α (SDF1α), also named CXCL12, is a chemokine first described in the bone marrow, which interacts with two receptors, CXCR4 and CXCR7 [13,14,15,16,17]. CXCL12/CXCR4 plays a cardinal role in the immune system as it regulates the development of T and B lymphocytes, the generation of memory T cells, and the maintenance of mature lymphocytes, and enhances the inflammatory response in a variety of settings [18,19,20].

In the mature brain, CXCL12 and its receptor CXCR4 are expressed in endothelial cells, glia, and neurons [21,22,23,24,25]. Regarding pathological conditions, CXCL12 is up-regulated in neurons and endothelia in the area of penumbra following brain ischemia, thus facilitating angiogenesis, neurogenesis, and synaptic transmission [26,27]. CXCL12 and CXCR7, but not CXCR4, are also increased in neurons and astrocytes in the periphery of human infarcts [28]. This chemokine also promotes the proliferation of radial glia and migration of neuroblasts along the corpus callosum following traumatic brain injury [29,30,31]. CXCL12 acting via CXCR4 facilitates the recruitment of progenitor cells to lesion sites and has a role in the regeneration of the nervous system in response to diverse injuries [20,32].

Previous studies have shown increased CXCL12 protein levels in the CSF in MS, and they can remain stable or increase over time [33,34,35,36,37]. Demyelination in MS and experimental autoimmune encephalitis (EAE) has been associated with the up-regulation of CXCL12, as a part of the damaging inflammatory response [38]. CXCL12 in MS is expressed in endothelial cells and astrocytes surrounding plaques [35,39]. However, the inhibition of CXCR4 activation during the induction of EAE increases the severity of the disease [40]. CXCL12 and CXCR4 are also co-expressed in oligodendrocyte precursor cells (OPCs) in the recovery phase in EAE, thus facilitating the capacity for repair following myelin and axonal damage [41]. This is in line with previous observations in cuprizone-induced demyelination animal models in which CXCR4 inhibition or silencing of CXCR4 mRNA expression impairs the differentiation of OPCs, resulting in failed remyelination [42]. Treatment with anti-CXCL12 blood serum decreases proliferation and migration of implanted neural stem cells or neural precursor cells, and impairs remyelination in a murine model of hepatitis virus-induced demyelination; moreover, blocking CXCR4 with the antagonist AMD3100 decreases the differentiation ability of implanted NSCs and NPCs [43]. CXCL12 expression is strongly increased in active remyelinating and inactive demyelinated lesions in MS [35,44]. Yet, the interpretation of this observation is not uniform. CXCL12 localization in blood vessels indicates a possible role in leukocyte extravasation, and it suggests a contribution to axonal damage [35]. On the other hand, the presence of CXCL12 in remyelinating lesions suggests a role for the chemokine in oligodendroglionesis and myelin repair in MS [44]. In this line, it is worth noting that the intranasal delivery of SDF-1α-preconditioned bone marrow mesenchymal cells improves remyelination in a cuprizone-induced mouse model of multiple sclerosis [45]. CXCL12 preconditioning increases CXCR4 expression, and the expression of oligodendrocyte lineage transcription factor-2 (Olig-2), enhances myelination, reduces the expression of glial fibrillary acidic protein and Iba-1, and improves spatial learning and memory [45].

In the human spinal cord CXCL12 is mainly expressed in motor neurons; CXCL12 mRNA levels are increased in the anterior horn of the lumbar spinal cord in sALS, and CXCL12 protein expression is found in the remaining motor neurons in sALS, and in a few glial cells, although the total levels of the protein in the spinal cord homogenates in sALS are within the range of control levels. CXCR4 mRNA, but not CXCR7 mRNA, is up-regulated in the spinal cord in sALS; CXCR4 protein is localized in axonal spheroids and degenerating motor neurons in the anterior horn of the spinal cord, and in a subset of oligodendroglia-like cells in the pyramidal tracts in sALS, while CXCR7 localizes in motor neurons in control and sALS, and reactive astrocytes in the pyramidal tracts in sALS.

CXCR4 was previously studied in ALS mouse models. CXCR4 undergoes early autocrine and proteostatic deregulation, and intracellular sequestration and aggregation as a result of Ranbp2 loss in mouse Thy1 motoneurons which causes ALS syndromes with hypoactivity followed by hindlimb paralysis, respiratory distress, and, ultimately, death [46]. CXCR4 is reduced in immortalized glial restricted precursors (GRIPs) derived from mouse E11.5 neural tubes of wild-type and SOD1(G93A) mutant mice. Subsequently, SOD1(G93A) GRIPs are unable to respond to SDF1alpha to activate ERK1/2 enzymes and the transcription factor CREB, thus impairing signaling through CXCR4 [47]. Targeting CXCR4 might have clinical implications in sALS. Chronic administration of AMD3100, an antagonist of CXCR4 to SOD1(G93A) mice improved microglial pathology, decreased pro-inflammatory cytokines in spinal cords, decreased blood-spinal cord barrier permeability by increasing tight junction proteins levels, and increased the motor neurons count in the lamina X area of the spinal cord. Moreover, AMD3100 led to a significant extension in mouse lifespan and improved motor function and weight loss [48].

In the present series, a significant increase in CXCL12 levels is found in the CSF of MS patients, thus being in line with previous studies in MS. However, CXCL12 protein levels are not elevated in AD, SMA, and FTD assessed for comparative study. Since CXCL12 levels in the CSF are also increased in MS, CXCL12 cannot be considered as a specific marker of sALS. However, the interpretation of biochemical data is always made within the context of the clinical manifestations of individual patients. NF-L and YKL40 protein levels in the CSF are used as biomarkers of disease prognosis in sALS. The present findings show that the assessment of CXCL12 levels in CSF may be used as a complementary diagnostic biomarker in sALS. Since CXCL12 levels at the first stages of the disease do not correlate with the age of onset, first clinical symptoms, or progression of the disease, CXCL1 cannot be considered a biomarker of prognosis. CXCL12 levels correlate positively with YKL40 but not with NF-L protein levels in sALS. This suggests that CXCL12 is a molecule best linked to the neuroinflammatory response rather than to neuronal damage; in this line CXCL12-immunoreactive glial cells are observed in sALS but not in controls. Yet, the localization of CXCR4 in axonal spheroids and degenerating motor neurons, and in a subset of oligodendrial cells and/or precursors in the pyramidal tracts, together with the presence of CXCR7 in motor neurons and reactive astrocytes in the pyramidal tracts, suggests a more complex scenario for CXCL12 involving neuron/axon preservation, oligodendroglial activation, and astrocyte signaling in sALS.

## 4. Materials and Methods

### 4.1. CSF Collection

Cerebrospinal fluid (CSF) was prospectively collected from patients undergoing lumbar puncture due to clinical suspicion of motor neuron disease at the functional unit of amyotrophic lateral sclerosis (UFELA) of the Neurology Service of the Bellvitge University Hospital. sALS was diagnosed according to updated El Escorial criteria [49,50]. Patients were evaluated clinically according to the main signs at onset (spinal, bulbar, and respiratory) and categorized according to disease progression as fast, expected, and slow progression depending on the survival. Fast progression included patients who survived less than 3 years, normal progression between 3 and 5 years, and slow progression for those still alive after 5 years. The ALS Functional Rating Scale Revised (ALS-FRS-R, version May 2015) was used in every case through the clinical course of the disease. At the time of obtaining the CSF, patients with spinal onset rated 4 or 3 in the ALS-FRS-R corresponding subscales; patients with bulbar onset rated 3 or 2 in the capacities of speech, salivation, and swallowing; patients with respiratory onset had dyspnea at rest. No patients needed respiratory support or gastrostomy.

CSF was obtained at the first visit of diagnosis. The results of routine analysis of CSF (leukocytes, proteins) were normal; red blood cells were not present in any case. In these patients, 1.5 ± 0.5 mL of CSF was collected in polypropylene tubes. CSF was centrifuged at 3000 rpm for 15 min at room temperature. The supernatant was collected and aliquoted in volumes of 220 μL and stored at −80 °C until use. Cases are summarized in Table 2. Cohort 1 includes CSF from 43 sALS cases and 36 healthy donors following the protocols for the use of biological samples for research. CSF in controls was obtained from patients at the need for knee surgical procedures under spinal anesthesia. All the cases of the first cohort were used for ELISA, but 15 sALS cases and 15 controls chosen at random from this cohort were used for SWATH-based proteomic analysis. A second cohort (validation cohort) of 44 healthy control and 65 sALS cases was analyzed for validation purposes following the same technical and ethical protocols. In the first cohort of sALS cases, the mean time to diagnosis was 10 months (between 2 and 43 months); the average time collection sample 10 months (between 2 and 38 months) after the onset of symptoms. The mean time of follow-up was 11 months (between 2 and 49 months). Sixteen patients died at the time of writing this paper; the mean survival time was 23 months (between 5 and 66 months) after the onset of symptoms. In the second cohort, the mean time to diagnosis was 11 months (between 2 and 49 months); the average time collection sample 11 months (between 2 and 65 months). The patients were followed for 19 months (between 1 and 50 months). The mean survival time of deceased patients was 29 months (between 4 and 84 months) after the onset of symptoms. The mean survival time of patients alive was 45 months (between 29 and 79 months) from disease onset.

Four additional groups of different pathological conditions were also obtained from different centers and units for comparative purposes: 30 multiple sclerosis (MS) cases and 13 adult spinal muscular atrophy patients, 6 type II (SMA-II) and 7 type III recruited from the Neurology Service of the Bellvitge University Hospital, 19 Alzheimer’s disease (AD) cases recruited at the Clinical Dementia Center and the National Reference Center for Creutzfeldt-Jakob disease surveillance McKhann, Göttingen University Medical Center, Göttingen, (Germany), and 39 cases with behavioral variant FTD (bvFTD) diagnosed according to the International Behavioral Variant FTD Criteria Consortium for bvFTD [51] obtained from the Dementia Clinic, Neurology Department of Coimbra-University Hospital (Portugal). None of them had motor neuron disease. sALS cases in these series did suffer from dementia. sALS cases in the first and second cohorts did not carry *C9orf72* expansions, *SOD1*, and *TARDBP* mutations; other genes were not assessed. Cases are shown in Table 2. Curiously, females were more common than males in the first cohort (29 and 14, respectively) for unknown reasons. Cases and controls had not suffered from infection or inflammatory diseases at the time of sampling. CSF samples were obtained according to the Declaration of Helsinki, and following informed consent and approval by the Clinical Research Ethics Committee (CEIC) of the Bellvitge University Hospital, or following the protocols accepted by the respective institutions (Reference numbers: 11/11/93 and 9/06/08, Universitätsmedizin Göttingen, Germany and HUC-43-09, University of Coimbra, Coimbra, Portugal).

### 4.2. Proteomics Analysis

#### 4.2.1. Building an MS/MS Library for SWATH Analysis

CSF samples from control and ALS cases (n = 15, in each group) were mixed with lysis buffer containing 7 M urea, 2 M thiourea, and 50 mM DTT (*v*/*v*; 1:1). As input for generating the SWATH-MS assay library, a pooled sample derived from all the subjects was prepared. Protein concentration was measured using the Bradford assay kit (Bio-rad). Pooled protein sample (20 μg) was diluted in Laemmli sample buffer and loaded into a 0.75 mm thick polyacrylamide gel with a 4% stacking gel cast over a 12.5% resolving gel. The total gel was stained with Coomassie brilliant blue, and 12 identical slides from the pooled sample were excised from the gel and transferred into 1.5 mL Eppendorf tubes. In-gel protein enzymatic cleavage, label-free LC-MS/MS, and library generation were performed as previously described [52].

#### 4.2.2. Quantitative Proteomics with SWATH Analysis

In-solution protein digestion (20 μg) from each sample was performed as described elsewhere [53].

### 4.3. LC-SWATH-MS Analysis

For SWATH-MS-based experiments, the instrument (Sciex TripleTOF 5600+) was configured as described [54]. Briefly, the mass spectrometer was operated in a looped-product ion mode. In this mode, the instrument specifically tuned to allow a quadrupole resolution of Da/mass selection. The stability of the mass selection was maintained by the operation of the Radio Frequency (RF) and Direct Current (DC) voltages on the isolation quadrupole in an independent manner. Using an isolation width of 16 Da (15 Da of optimal ion transmission efficiency and 1 Da for the window overlap), a set of 37 overlapping windows was constructed covering the mass range 450–1000 Da. Consecutive swaths were needed with some precursor isolation window overlap to ensure the transfer of the complete isotopic pattern of any given precursor ion to at least one isolation window, thereby maintaining an optimal correlation between parent and fragment isotopes peaks at an LC time point. In this way, 1 μL of each sample was loaded onto a trap column (Thermo Scientific Barcelona,m Spain) 0.1 × 50 mm, particle size 5 μm, and pore size 100 Å), and desalted with 0.1% TFA at 3 μL/min for 10 min. The peptides were loaded onto an analytical column (Thermo Scientific 0.075 × 250 mm, particle size 3 μm, and pore size 100 Å) equilibrated in 2% acetonitrile 0.1% formic acid (FA). Peptide elution was carried out with a linear gradient of 2 to 40% B in 120 min (mobile phases A: 100% water 0.1% FA, and B: 100% acetonitrile 0.1% FA) at a flow rate of 300 nL/min. Eluted peptides were infused in the spectrometer TripleTOF+ 5600 (Sciex, Concord, ON, Canada). The TripleTOF was operated in swath mode, in which a 0.050 s TOF MS scan from 350 to 1250 m/z was performed, followed by 0.080 s production scans from 230 to 1800 m/z on the 37 defined windows (3.05 s/cycle). The collision energy was set to optimum for 2^+^ ion at the center of each SWATH block with 15 eV collision energy spread. The mass spectrometer was always operated in high sensitivity mode.

### 4.4. Protein Identification and Quantification

The resulting ProteinPilot group file from library generation was loaded into PeakView^®^ (v2.1, Sciex) Concord, Canada) and peaks from SWATH runs were extracted with a threshold of 99% confidence (Unused Score ≥ 1.3), and a false discovery rate (FDR) lower than 1%. MS/MS spectra of the assigned peptides were extracted by ProteinPilot, and only the proteins that fulfilled the following criteria were validated: (1) peptide mass tolerance lower than 10 ppm, (2) 99% confidence level in peptide identification, and (3) complete b/y ion series found in the MS/MS spectrum. The quantitative data obtained with PeakView^®^ were analyzed using Perseus software [55] for statistical analysis and data visualization. Two-sample *t*-test based on permutation-based FDR statistics was applied (250 permutations); proteins with a *p*-value lower than 0.05, and absolute fold-change of <0.77 (down-regulation) or >1.3 (up-regulation) in linear scale were considered significantly differentially expressed and used for further evaluation.

### 4.5. CSF Analysis

C-X-C motif chemokine ligand 12 (CXCL12) was quantified using the Human CXCL12/SDF-1α Quantikine ELISA Kit from R&D Systems according to the manufacturer’s instructions (R&D Systems, Inc., Minneapolis, MN, USA). AAAS (Aladin WD repeat nucleoporin) and proteasomal ATPase associated factor 1 (PAAF1) were quantified using the Human AAAS (Aladin) ELISA kit and Human PAAF1 ELISA kit, respectively, from FineTest according to the manufacturer’s instructions (Wuhan Fine Biotech Co, Hubei, China). However, CSF samples for AAAS quantification were diluted 1:3. Syntaxin-12 was quantified using Human Syntaxin 12 (STX12) ELISA Kit from MyBioSource following the manufacturer’s instructions (MyBiosource, San Diego, CA, USA). S100 calcium-binding protein A6 (S100A6) levels were quantified using CircuLex S100A6 ELISA Kit according to the manufacturer’s instructions (MBL International Corp, Woburn, MA, USA). Test performers were blinded to clinical information. Protein candidates were quantified and evaluated in the initial cohort of 36 control and 43 sALS cases.

### 4.6. Tissue Samples

Post-mortem samples were obtained from the Institute of Neuropathology HUB-ICO-IDIBELL Biobank following the guidelines of Spanish legislation on this matter and the approval of the CEIC of the Bellvitge University Hospital. The post-mortem interval between death and tissue processing was between 2 h and 17 h. Tissue processing was carried out as detailed elsewhere [56]. All cases met the neuropathological criteria for classical ALS regarding the involvement of the motor cortex, pyramidal tracts, and selected motor nuclei of the cranial nerves and anterior horn of the spinal cord [57]. Patients with associated pathology were excluded. Age-matched control cases had not suffered from neurologic or psychiatric diseases and did not have abnormalities in the neuropathological examination. A summary of sALS and control cases is shown in Table 3.

### 4.7. RNA Extraction and RT-qPCR Validation

Frozen samples of the anterior horn of the lumbar spinal cord (n = 22 sALS and n = 17 controls) were obtained for RNA extraction using RNeasy Mini Kit following the instructions of the supplier (Qiagen^®^ GmbH, Hilden, Germany). RNA integrity and 28S/18S ratios were determined with the Agilent Bioanalyzer (Agilent Technologies Inc, Santa Clara, CA, USA) to assess RNA quality, and the RNA concentration was evaluated using a NanoDrop™ Spectrophotometer (Thermo Fisher Scientific, Barcelona, Spain) (Table 3). Complementary DNA (cDNA) preparation used High-Capacity cDNA Reverse Transcription kit (Applied Biosystems, Foster City, CA, USA) following the protocol provided by the supplier. Parallel reactions for each RNA sample were run in the absence of MultiScribe Reverse Transcriptase to assess the lack of contamination of genomic DNA. TaqMan RT-qPCR assays were performed in duplicate for each gene on cDNA samples in 384-well optical plates using an ABI Prism 7900 Sequence Detection system (Applied Biosystems, Life Technologies, Waltham, MA, USA). For each 10 μL TaqMan reaction, 4.5 μL cDNA was mixed with 0.5 μL 20× TaqMan Gene Expression Assays and 5 μL of 2× TaqMan Universal PCR Master Mix (Applied Biosystems). Analyzed genes included the following TaqMan probes: CXCL12/SDF1α (Hs03676656_mH), C-X-C motif chemokine receptor 4 (CXCR4) (Hs00607978_s1), atypical chemokine receptor 3 (ACKR3/CXCR7) (Hs00664172_s1), allograft inflammatory factor (AIF1) (Hs00741549_g1) coding for IBA1, glial fibrillary acidic protein (GFAP) (Hs00909233_m1), CD68 (Hs02836816_g1) and chitinase 3 like 1 (CHI3L1/YKL40) (Hs01072228_m1). The mean value of one house-keeping gene, hypoxanthine-guanine phosphoribosyltransferase 1 (HPRT1), was used as the internal control for normalization [58]. The selection of HPRT1 as a house-keeping gene was due to our previous experience noting that other markers such as β-actin, tubulin, β-glucuronidase (GUS), superoxide dismutase 1 (SOD1), and metalloproteinase domain 22 (ADAM22) mRNAs had disparate expression in the human post-mortem control spinal cord. The parameters of the reactions were 50 °C for 2 min, 95 °C for 10 min, and 40 cycles of 95 °C for 15 s and 60 °C for 1 min. Finally, the capture of all TaqMan PCR data was with the Sequence Detection Software (SDS version 2.2.2, Applied Biosystems). The double-delta cycle threshold (ΔΔCT) method was used to analyze the data: the results from the student’s *t*-test. The significance level was set at * *p* < 0.05, ** *p* < 0.01 and *** *p* < 0.001, and tendencies at # < 0.1.

### 4.8. Gel Electrophoresis and Immunoblotting

Frozen samples of the anterior horn of the lumbar spinal cord (n = 7 sALS, n = 7 controls) were homogenized in RIPA lysis buffer composed of 50 mM Tris/HCl buffer, pH 7.4 containing 2 mM EDTA, 0.2% Nonidet P-40, 1 mM PMSF, protease, and phosphatase inhibitor cocktail (Roche Molecular Systems, USA). The homogenates were centrifuged for 20 min at 12,000 rpm. Protein concentration was determined with the BCA method (Thermo Scientific). Equal amounts of protein (12 μg) for each sample were loaded and separated by electrophoresis on 10–15% sodium dodecyl sulfate-polyacrylamide gel electrophoresis (SDS-PAGE) gels and then transferred onto nitrocellulose membranes (Amersham, Freiburg, GE). Non-specific bindings were blocked by incubation in 3% albumin in PBS containing 0.2% Tween for 1 h at room temperature. After washing, the membranes were incubated overnight at 4 °C with antibodies against CXCL12 (dilution 1:250, rabbit polyclonal, Abcam, Cambridge, UK). Protein loading was monitored using antibodies against β-actin (42 kDa, 1:30,000, Sigma, Barcelona, Spain). Membranes were incubated for 1 h with appropriate HRP-conjugated secondary antibodies (1:2000, Dako); the immunoreaction was revealed with a chemiluminescence reagent (ECL, Amersham). Densitometric quantification was carried out with the ImageLab v4.5.2 software (BioRad), using β-actin for normalization.

### 4.9. Immunohistochemistry

De-waxed sections, 4μm thick, of the lumbar spinal cord from control (n = 10) and sALS (n = 10) cases of those listed in Table 3 were processed in parallel for immunohistochemistry. Endogenous peroxidases were blocked by incubation in 10% methanol-1% H_2_O_2_ for 15 min followed by 3% normal horse serum. Then the sections were incubated at 4 °C overnight with one of the primary rabbit polyclonal antibodies: anti-CXCL12/SDF1 (Abcam, Cambridge, UK) diluted 1:100, anti-CXCR7 (Abcam, Cambridge, UK) diluted 1:100, and anti-CXCR4 (Abcam, Cambridge, UK) diluted 1:100. Following incubation with the primary antibody, the sections were incubated with EnVision + system peroxidase (Dako, Agilent, Santa Clara, CA, USA) for 30 min at room temperature. The peroxidase reaction was visualized with diaminobenzidine and H_2_O_2_. Control of the immunostaining included omission of the primary antibody; no signal was obtained following incubation with only the secondary antibody. Sections were lightly counter-stained with hematoxylin.

### 4.10. Statistical Analysis

The normality of distribution was analyzed with the Kolmogorov-Smirnov test. The unpaired Student’s *t*-test was used to compare two groups when values followed the normal distribution. Receiver operating characteristic (ROC) curves and derived area under the curve (AUC) were calculated. Best cut-off value, sensitivity, and specificity were estimated based on the Youden index (point on an ROC curve providing the best balance of sensitivity and specificity) [59]. ROC curve analyses were performed using MedCalc Software (MedCalc Software Ltd., Ostend Belgium, Belgium) and ROC curve comparisons with DeLong’s method [60]. Pearson’s correlation coefficients were used to assess associations between continuous biomarker levels and clinical and demographic data. Significance levels were set at * *p* < 0.05, ** *p* < 0.01, and *** *p* < 0.001. The statistical analysis of CSF protein data between groups was carried out using a one-way analysis of variance (ANOVA) followed by Tukey post-test using the SPSS software (IBM Corp. Released 2013, IBM-SPSS Statistics for Windows, Version 21.0., Armonk, NY, USA). Graphic design was performed with GraphPad Prism version 5.01 (La Jolla, CA, USA). Outliers were detected using the GraphPad software QuickCalcs (*p* < 0.05). The data were expressed as mean ± SEM; significance levels were set at * *p* < 0.05, ** *p* < 0.01, and *** *p* < 0.001, and tendencies at # *p* < 0.1.

## Figures and Tables

**Figure 1 ijms-21-08680-f001:**
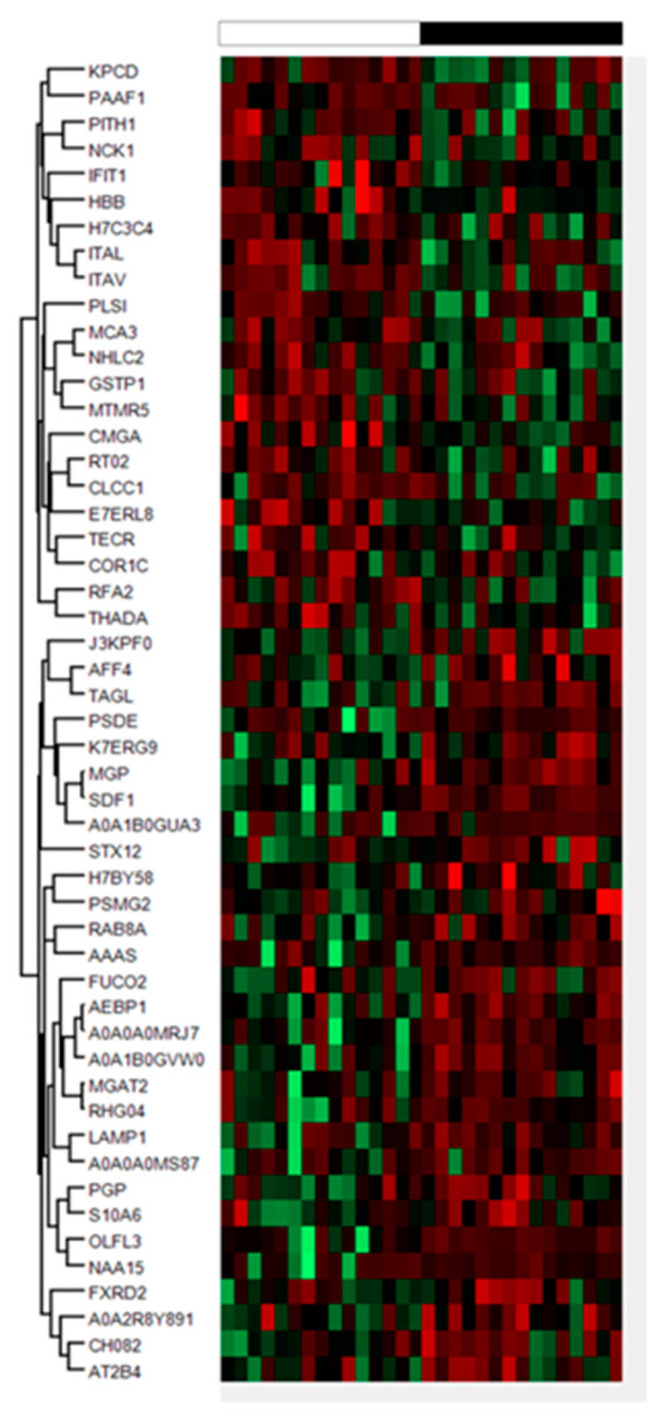
Hierarchical clustering heat-map of expression intensities of proteins reflects differential protein profiles in cerebrospinal fluid (CSF) of sporadic amyotrophic lateral sclerosis (sALS) cases (*n* = 15, black) compared with controls (n = 15, white). Differences between groups are considered statistically significant at *p*-value ≤ 0.05.

**Figure 2 ijms-21-08680-f002:**
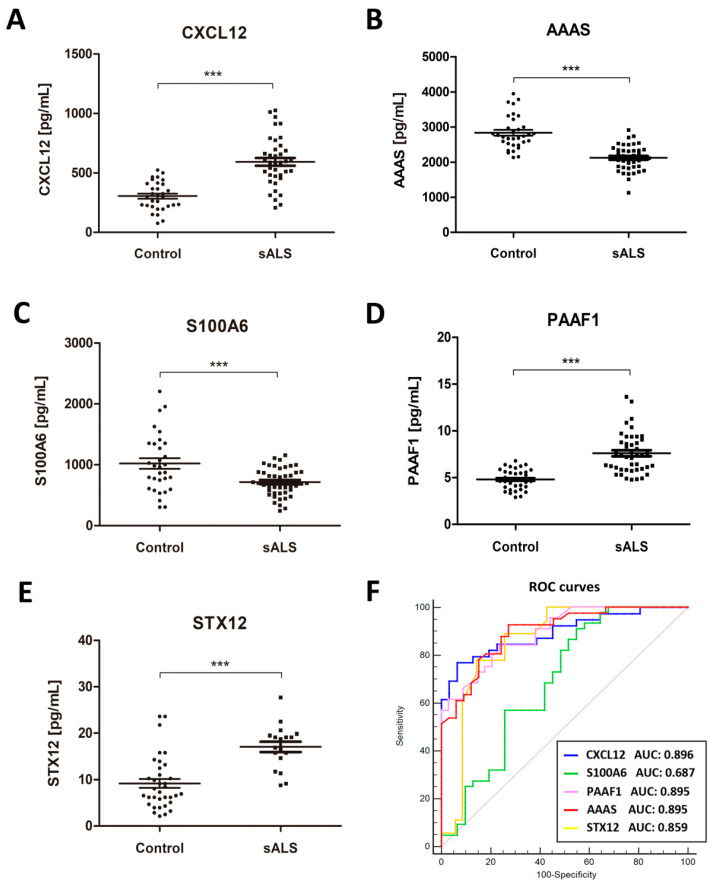
Quantification of CXCL12 (**A**), AAAS (**B**), S100A6 (**C**) PAAF1 (**D**), and STX12 (**E**) protein levels in the CSF in sALS (n = 43) and control (n = 36) cases using ELISA. Mean ± SEM is represented in the graphs. Differences between sALS patients and controls were assessed with a *t*-test. Significance was set at *** *p* < 0.001. (**F**) receiver operating characteristic (ROC) curves of CXCL12, AAAS, S100A6, STX12, and PAAF1 quantification in the differential diagnosis of sALS compared with controls. AUC values, corresponding to the area under ROC curves, and 95% confidence intervals are indicated.

**Figure 3 ijms-21-08680-f003:**
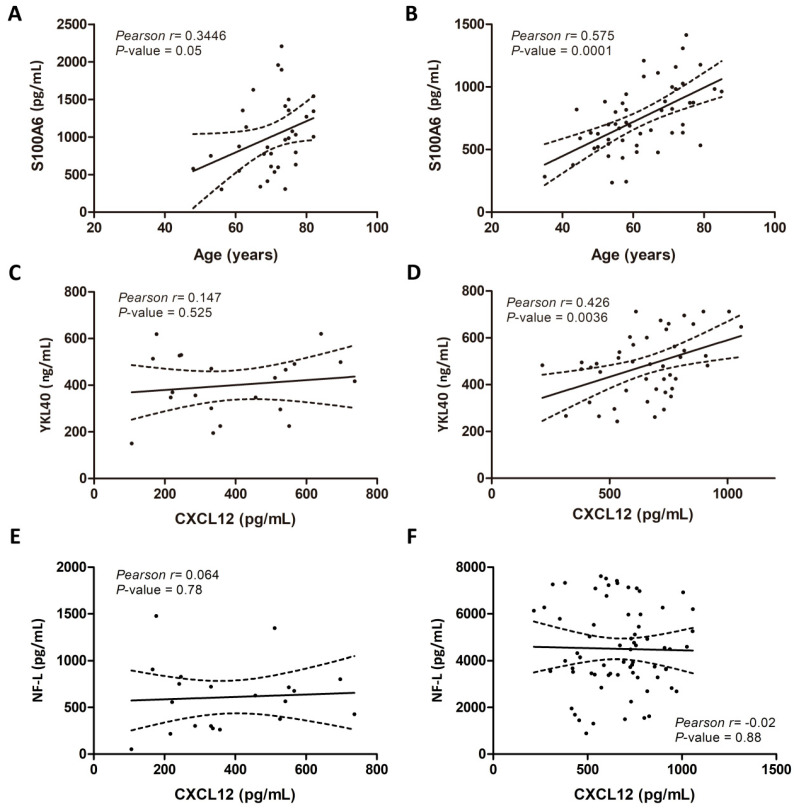
Correlation between clinical parameters and biomarker levels in the CSF. Left panels correspond to control case correlations (**A**,**C**,**E**) and right panels correspond to sALS case correlations (**B**,**D**,**F**). S100A6 levels in the CSF of control (**A**) and sALS cases (**B**) correlate with age, whereas CXCL12 levels correlate with YKL40 levels in sALS cases (**D**) but not control cases (**C**). However, CXCL12 levels do not correlate with NF-L levels in control (**E**) or sALS cases (**F**). Positive correlations between variables were analyzed using Pearson’s correlation test; statistical significance was established to two-tailed *p*-values < 0.05.

**Figure 4 ijms-21-08680-f004:**
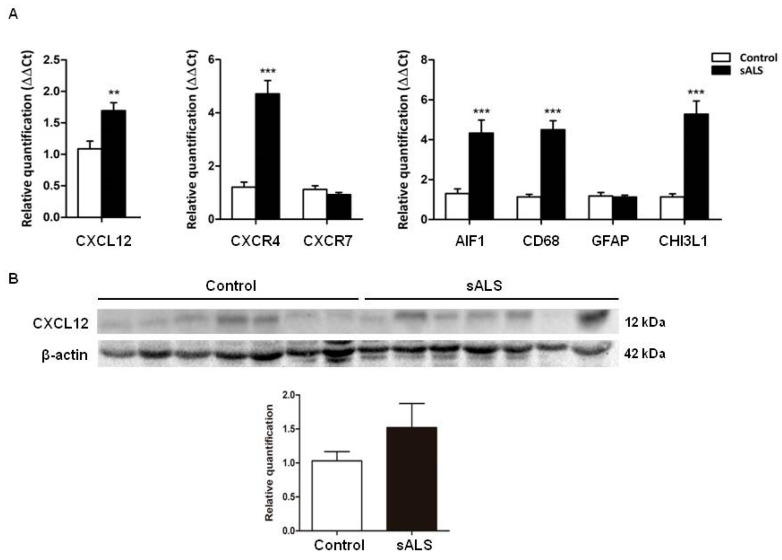
(**A**) CXCL12, CXCR4, CHI3L1, CD68, and AIF1 mRNA expression is significantly increased in the anterior horn of the lumbar spinal cord in sALS when compared with controls; CXCR7 and GFAP mRNA expression is not modified in sALS when compared with controls. (**B**) Western blot analysis of anterior horn lumbar spinal horn lysates shows a trend of increased CXCL12 protein levels in sALS when compared with controls, but the difference from the control was not statistically significant. β-actin is used for normalization. Graphical representation of Western blots expresses fold changes in sALS relative to control cases. Mean ± SEM is represented in the graphs. Differences between sALS patients and controls were assessed with a *t*-test. Significance was set at ** *p* < 0.01 and *** *p* < 0.001.

**Figure 5 ijms-21-08680-f005:**
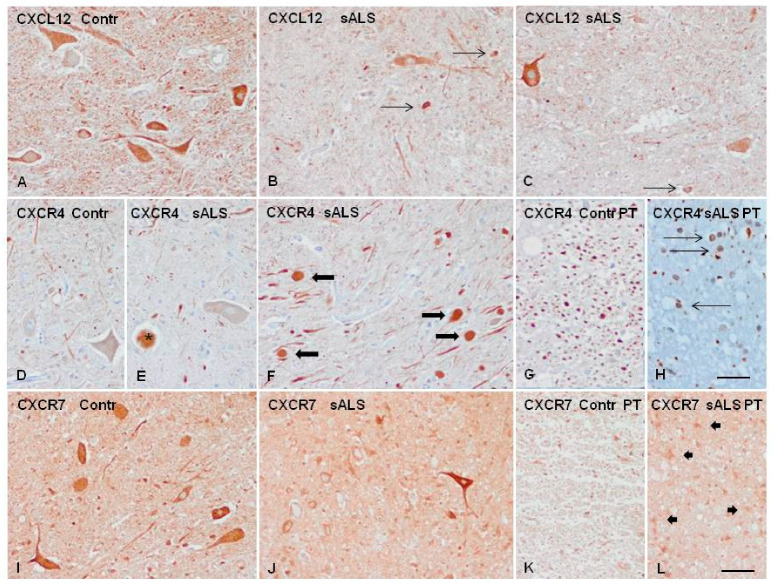
Immunohistochemistry to CXCL12, CXCR4, and CXCR7 in control (Contr) and sALS spinal cord. (**A**–**C**) CXCL12 immunoreactivity is found in motor neurons of the spinal cord in control and sALS, and in a few glial cells in sALS (thin arrows). (**D**) CXCR4 immunoreactivity mainly occurs in axons in controls. (**E**,**F**) CXCR4 immunoreactivity in sALS is observed in degenerating neurons (asterisk in (**E**)) and in axonal ballooning (thick arrows in (**F**)) in the anterior horn of the spinal cord. (**G**) CXCR4 also decorates axons in the pyramidal tracts (PT) in controls, and (**H**) in a subset of oligodendroglia-like cells in the axon-deprived PT in sALS (thin arrows). (**I**,**J**) CXCR7 immunostaining decorates the cytoplasm of motor neurons in control and sALS. (**K**,**L**) CXCR7 immunostaining is weak in the pyramidal tracts in controls (**K**) but moderate in reactive astrocytes in sALS ((**L**), thick arrows). Paraffin sections are lightly counterstained with hematoxylin, bar = 50 μm.

**Figure 6 ijms-21-08680-f006:**
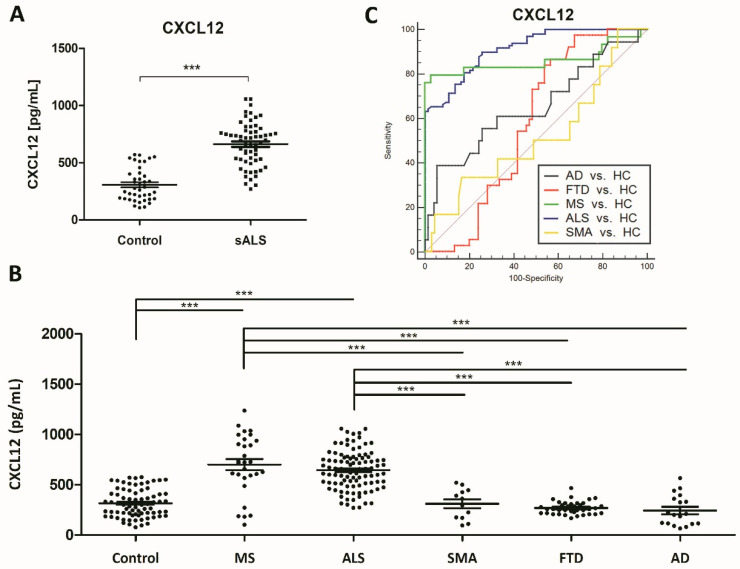
(**A**) Validation of CXCL12 CSF levels in the second cohort of sALS and control cases. Mean ± SEM is represented in the graphs. Differences between sALS patients and controls were assessed with a *t*-test. Significance was set at *** *p* < 0.001. (**B**) CXCL12 levels in different neurodegenerative disorders with cognitive decline or motor involvement. CXCL12 levels are significantly increased in sALS (the combination of the first and second cohort) and MS cases, whereas CXCL12 protein levels are not significantly altered in the CSF in AD, SMA, and FTD cases, compared with controls. Mean ± SEM is represented in the graphs. Differences between sALS patients and controls were assessed with a one-way ANOVA test. Significance was set at *** *p* < 0.001. (**C**) ROC curves of CXCL12 in Alzheimer’s disease (AD), spinal muscular atrophy III (SMA), multiple sclerosis (MS), and behavioral variant frontotemporal dementia (FTD).

**Table 1 ijms-21-08680-t001:** List of differentially expressed proteins in the CSF of sALS cases compared with age-matched controls using SWATH acquisition on liquid chromatography-tandem mass spectrometry (LC–MS/MS).

Gene		Accesion	Peptides	%CV	FC	*p*-Value
HBB	hemoglobin subunit beta	P68871	7	105.28	0.37	0.04
CHGA	chromogranin A	P10645	6	72.40	0.55	0.00
E7ERL8	neurexin 1	E7ERL8	5	63.37	0.66	0.02
CLCC1	chloride channel CLIC like 1	Q96S66	3	33.26	0.64	0.04
K7ERG9	complement factor D	K7ERG9	3	40.73	1.41	0.02
TECR	trans-2,3-enoyl-CoA reductase	Q9NZ01	2	42.33	0.67	0.04
ARHGAP4	Rho GTPase activating protein	P98171	2	72.50	1.32	0.02
FUCA2	alpha-L-fucosidase 2	Q9BTY2	2	56.10	1.57	0.04
A0A0A0MRJ7	-	A0A0A0MRJ7	2	66.71	2.12	0.00
NCK1	NECK adaptor protein 1	P16333	1	102.16	0.35	0.04
MTMR5	SBF1 (MTMR5) SET binding factor 1	O95248	1	82.74	0.36	0.02
COR1C	coronin-1C	Q9ULV4	1	98.19	0.40	0.05
ITAV	integrin, alpha 5	P06756	1	16.49	0.41	0.01
PAAF1	proteasomal ATPase associated factor 1	Q9BRP4	1	83.56	0.43	0.03
ITAL	toll-like receptor 4	P20701	1	78.52	0.43	0.01
MCA3	eukaryotic translation elongation factor 1 epsilon-1	O43324	1	100.07	0.45	0.04
KPCD	protein kinase C delta type i	Q05655	1	41.25	0.46	0.03
THADA	THADA armadillo repeat containing	Q6YHU6	1	46.64	0.48	0.01
NHLC2	NHL repeat-containing protein 2	Q8NBF2	1	73.60	0.49	0.03
PLSI	plastin 1	Q14651	1	25.89	0.54	0.03
RFA2	replication protein A2	P15927	1	93.57	0.54	0.01
MRPS2	mitochondrial ribosomal protein S2	Q9Y399	1	61.86	0.56	0.02
GSTP1	glutathione S-transferase pi1	P09211	1	30.22	0.57	0.04
PITH1	PITH domain containing 1	Q9GZP4	1	60.84	0.60	0.02
IFIT1	interferon induced protein with tetratricopeptide repeats 1	P09914	1	41.39	0.63	0.05
H7C3C4	-	H7C3C4	1	32.17	0.70	0.04
MGAT2	alpha-1,6-mannosyl-glycoprotein 2-beta-N-acetylglucosaminyltransferase	Q10469	1	34.5	1.39	0.02
AFF4	AF4/FMR2 familiy member 4	Q9UHB7	1	37.83	1.39	0.03
PGP	phosphoglycolate phosphatase	A6NDG6	1	20.82	1.45	0.04
AEBP1	AE binding protein 1	Q8IUX7	1	39.01	1.57	0.00
PSMG2	proteasome assembly chaperone 2	Q969U7	1	40.08	1.60	0.03
RAB8A	RAB8A, member RAS oncogene family	P61006	1	77.36	1.66	0.01
S10A6	S100 calcium biding protein A6	P06703	1	64.07	1.67	0.03
A0A2R8Y891	-	A0A2R8Y891	1	46.81	1.91	0.02
LAMP1	lysosomal associated membrane protein 1	P11279	1	99.83	1.92	0.02
FXRD2	FAD-dependent oxidoreductase domain containing 2	Q8IWF2	1	56.92	1.95	0.05
SDF1	C-X-C motif chemokine ligand 12	P48061	1	39.23	1.97	0.00
A0A0A0MS87	-	A0A0A0MS87	1	67.51	1.98	0.04
MGP	matrix Gla protein	P08493	1	28.51	2.00	0.00
A0A1B0GVW0	-	A0A1B0GVW0	1	60.77	2.01	0.00
STX12	syntaxin 12	Q86Y82	1	60.13	2.01	0.04
AT2B4	ATPase plasma membrane Ca2+ transportin 4	P23634	1	59.88	2.04	0.04
PSDE	proteasome 26S subunit, non-ATPase 14	O00487	1	69.53	2.12	0.04
PCMT1	protein-l-isoaspartate (D-aspartate) O-methyltransferase	H7BY58	1	50.17	2.14	0.03
CH082		Q6P1X6	1	52.14	2.19	0.03
AAAS	aladin WD repeat nucleoporin	Q9NRG9	1	59.66	2.19	0.02
A0A1B0GUA3	-	A0A1B0GUA3	1	123.34	2.27	0.03
OLFL3	olfactomedin 3	Q9NRN5	1	62.69	2.57	0.00
TAGL	transgelin	Q01995	1	136.16	2.95	0.01
NAA15	N-alpha.-acetyltransferase 15, NatA auxiliary subunit	Q9BXJ9	1	130.87	3.26	0.04
J3KPF0	-	J3KPF0	1	114.46	3.66	0.00

**Table 2 ijms-21-08680-t002:** List of cases used for CSF studies including the number of cases (n), age (mean ± standard deviation), sex (female/male), and clinical form at disease onset and the number of cases; abbreviations: AD: Alzheimer’s disease, FTD: frontotemporal dementia, bvFTD: behavioral variant FTD, MS: multiple sclerosis, RRMS: relapsing-remitting MS, sALS: sporadic amyotrophic lateral sclerosis, SMA: spinal muscular atrophy type III, HC: healthy control. Disease subtype indicates the main symptom at onset; resp: respiratory.

Group	n	Age	Sex (f/m)	Disease Subtype
**Cohort 1**				
HC	36	67 ± 12.9	(21/15)	
sALS	43	61 ± 10.8	(29/14)	21 Spinal
				21 Bulbar
				1 Resp
**Cohort 2**				
HC	44	68 ± 09.7	(26/18)	
sALS	65	66 ± 12.3	(27/38)	43 Spinal
				21 Bulbar
				1 Resp
MS	30	41 ± 10.7	(22/8)	18 Acute-MS
				12 RRMS
SMA-III	13	31 ± 7.75	(6/7)	Type III
AD	19	64 ± 08.6	(14/5)	
FTD	39	66 ± 08.5	(18/21)	39 bvFTD

**Table 3 ijms-21-08680-t003:** Summary of the thirty-nine cases used for the study of the spinal cord: 17 controls and 22 sALS cases. Abbreviations: ALS: amyotrophic lateral sclerosis; F: female; M: male; PM: post-mortem delay (hours, minutes); SC: anterior horn of the spinal cord lumbar level; RIN: RNA integrity number; N/A: not available. Site of onset: main symptom at onset.

Case	Age	Gender	Diagnosis	PM Delay	Site of Onset	RIN
1	70	M	ALS	03 h 00 min	Resp.	7.00
2	77	M	ALS	04 h 30 min	N/A	7.50
3	83	F	ALS	15 h 15 min	N/A	7.20
4	56	F	ALS	03 h 45 min	N/A	8.10
5	56	M	ALS	10 h 50 min	N/A	6.60
6	76	M	ALS	12 h 40 min	Spinal	7.00
7	69	M	ALS	02 h 00 min	N/A	7.00
8	63	F	ALS	13 h 50 min	Bulbar	6.50
9	N/A	M	ALS	N/A	N/A	8.70
10	65	F	ALS	04 h 10 min	N/A	7.70
11	50	M	ALS	10 h 10 min	Spinal	5.30
12	71	M	ALS	03 h 25 min	N/A	8.10
13	54	M	ALS	04 h 50 min	Spinal	8.80
14	64	M	ALS	16 h 30 min	N/A	6.70
15	75	F	ALS	04 h 05 min	Bulbar	8.50
16	76	F	ALS	13 h 00 min	N/A	8.10
17	57	F	ALS	10 h 00 min	Bulbar	7.00
18	79	F	ALS	02 h 10 min	Bulbar	8.10
19	57	F	ALS	04 h 00 min	Bulbar	6.20
20	46	M	ALS	07 h 00 min	Spinal	7.00
21	69	F	ALS	17 h 00 min	Spinal	6.40
22	59	M	ALS	03 h 15 min	N/A	6.80
23	66	M	Control	14 h 00 min	-	5.00
24	46	M	Control	15 h 00 min	-	5.70
25	66	M	Control	05 h 00 min	-	5.40
26	77	F	Control	08 h 30 min	-	5.10
27	64	F	Control	05 h 00 min	-	7.00
28	60	F	Control	09 h 40 min	-	5.80
29	52	M	Control	03 h 00 min	-	5.00
30	67	M	Control	07 h 00 min	-	5.50
31	47	M	Control	04 h 55 min	-	5.60
32	64	F	Control	11 h 20 min	-	6.20
33	56	M	Control	07 h 10 min	-	6.10
34	71	F	Control	08 h 30 min	-	5.90
35	55	M	Control	09 h 45 min	-	5.30
36	75	M	Control	07 h 30 min	-	6.60
37	51	F	Control	04 h 00 min	-	6.30
38	59	M	Control	12 h 05 min	-	6.40
39	75	F	Control	10 h 30 min	-	5.20

## Data Availability

All data generated or analyzed during this study are included in this published article. Rough data are available upon reasonable request.

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
