# Peer review of "Increased C-X-C Motif Chemokine Ligand 12 Levels in Cerebrospinal Fluid as a Candidate Biomarker in Sporadic Amyotrophic Lateral Sclerosis"

_ijms, 2020, doi:10.3390/ijms21228680_

Round 1

Reviewer 1 Report

The manuscript by Pol Andrés-Benito et al. is on actual topic, with highly considered experimental design and with very extensive validation using clinical samples. It is the first study employing quantitative SWATH MS proteomic approach to search for ALS biomarkers in cerebrospinal fluid (previously published study involved blood plasma). I recommend the manuscript for publication in IJMS journal, provided that several minor revisions are provided.

Suggested minor revisions:

  1. Title: the title is too ambitious.

I suggest to change „biomarker“ to „candidate biomarker“ as the biomarker needs more elaborate evaluation (as described e.g. in Steven J. Skates et al., 2013).

I suggest to remove „new player“ from the title, as the function of CXCL12 in ALS was not studied in this paper.

I suggest to add to the title information „in cerebrospinal fluid“.

  1. Abstract, 1st sentence: consider to change the word „involving“ to „affecting“

  1. Abstract, 3rd paragraph: abbreviations AD, FTD, SMA, MS should be used as they were defined previously

  1. Results: add information, how many proteins were quantified in total by SWATH MS in all CSF samples and/or what what was the average number of quantified proteins in a single sample.

  1. Results: Figure 4B + line 186-188 in the text – western blot results should be more informatively presented and described.

The text (line 186-188) says, that there is no statistically significant difference between the CXCL12 level in anterior horns of control samples and ALS patients. However, the graph in Figure 4B shows higher CXCL12 level in tissue in ALS compared to controls, which is similar to CXCL12 protein in CSF (ELISA), as well as to CXCL12 mRNA level in spinal anterior horns (qPCR).

I suggest to mention, that there was a trend of increased CXCL12 protein expression in spinal cord tissue in ALS, but the difference from control was not statistically significant.

It is difficult to assess the western blot results, when only 2 control and 2 sALS samples are shown. Can you show western blot scans of all 7 analyzed controls and 7 sALS samples? Was the quantification performed on not overexposed images/films?

  1. Results: Figure 5 needs more detailed description or more clear organization. It should be labelled, which spinal cord areas are shown on panels (e.g. anterior horns, pyramidal tracts). CXCR4 staining in anterior horns in controls is not shown. The panel 5C looks to be captured at different magnification. How many samples in controls and sALS were studied by immunohistochemistry?

  1. Methods: Why was the mRNA expression normalized to just 1 reference gene? Normalization to 3 genes is broadly accepted.

  1. Discussion:

Add information that the CXCL12 receptor CDCR4 was previously studied in ALS mice models. Kyoung-In Cho et al. 2017 and Yongquan Luo et al., 2007 described impaired signalling through CXCR4 in ALS mouse models. Treatment of the SOD1(G93A) mice with CXCR4 antagonist extended the lifespan and improved motor functions (Inna Rabinovich-Nikitin et al., 2016).

  1. Typing errors:

Line 180 – add more details to the sentence „Significantly increased mRNA levels of CXCL12 (P=0.002).“ – what was compared?

Lines 197-198 – missing number of analyzed cases

Figure 4 legend: CXCR7 and GFAP mRNA expression is npot modified (should be „not modified“)

Author Response

We thank the reviewers for their comments and suggestionsReviewer 1Suggested minor revisions:1.Title: the title is too ambitious.I suggest to change „biomarker“ to„candidate biomarker“ as the biomarker needs more elaborate evaluation (as described e.g. in Steven J. Skates et al., 2013).CorrectedI suggest to remove „new player“from the title, as the function of CXCL12 in ALS was not studied in this paper.CorrectedI suggest to add to the title information „in cerebrospinal fluid“.Corrected2.Abstract, 1st sentence: consider to change the word „involving“ to „affecting“Corrected3.Abstract, 3rd paragraph: abbreviations AD, FTD, SMA, MS should be used as they were defined previouslyCorrected4.Results: add information, how many proteins were quantified in total by SWATH MS in all CSF samples and/or what was the average number of quantified proteins in a single sample.Added 14685.Results: Figure 4B + line 186-188 in thetext western blot results should be more informatively presented and described.The text (line 186-188) says, that there is no statistically significant difference between the CXCL12 level in anterior horns of control samples and ALS patients. However, the graph in Figure 4B shows higher CXCL12 level in tissue in ALS compared to controls, which is similar to CXCL12 protein in CSF (ELISA), as well as to CXCL12 mRNA level in spinal anterior horns (qPCR).I suggest to mention, that there was a trend of increased CXCL12 protein expression in spinal cord tissue in ALS, but the difference from control was not statistically significant.CorrectedIt is difficult to assess the western blot results, when only 2 control and 2 sALS samples are shown. Can you show western blot scans of all 7 analyzed controls and 7 sALS samples? Was the quantification performed on not overexposed images/films?Western blots scans of the 7 sALS and 7 controls are added in Figure 4B6.Results: Figure 5 needs more detailed description ormore clear organization. It should be labelled, which spinal cord areas are shown on panels (e.g. anterior horns, pyramidal tracts). CXCR4 staining in anterior horns in controls is not shown. The panel 5C looks to be captured at different magnification. How many samples in controls and sALS were studied by immunohistochemistry?Figure 5 has been modified following your suggestionsThe number of cases examined is now indicated in material and methods7.Methods: Why was the mRNA expression normalized to just 1 reference gene? Normalization to 3 genes is broadly accepted.We are very concerned about the optimal use of house-keeping genes for normalization. We have assessed several genes, but in the human post-mortem spinal cord most of them offer variegated individual expression levels in normal

subjects. We have explained this aspect in the new version in Material and methods 8.Discussion:Add information that the CXCL12 receptor CDCR4 was previously studied in ALS mice models. Kyoung-In Cho et al. 2017 and Yongquan Luo et al., 2007 described impaired signalling through CXCR4 in ALS mouse models. Treatment of the SOD1(G93A) mice with CXCR4 antagonist extended the lifespan and improved motor functions (Inna Rabinovich-Nikitin et al., 2016).We thank this useful information. These works are mentioned andthe corresponding references includedin the new version9.Typing errors:Line 180 add moredetails to the sentence „Significantly increased mRNA levels of CXCL12 (P=0.002).“ what was compared?CorrectedLines 197-198 missing number of analyzed casesCorrectedFigure 4 legend: CXCR7 and GFAP mRNA expression isnpotmodified (should be „not modified“)Corrected

Reviewer 2 Report

The manuscript presented by Andrés-Benito and coworkers is focused on the identification of putative biomarkers in the cerebrospinal fluid (CSF) of sALS patients at early disease stages compared with age-matched controls and with other neurodegenerative diseases including Alzheimer disease (AD), spinal muscular atrophy type III (SMA), frontotemporal dementia behavioral variant (FTD), and multiple sclerosis (MS). CXCL12 CSF levels was identified as a strong candidate to be used as a complementary diagnostic biomarker, in combination with CSF levels of the prognostic biomarkers YKL40 and NF-L in sALS. The findings obtained are relevant to the field of ALS biomarker discovery. 

Albeit the authors obtain a lack of correlation of CXCL12 CSF levels with NF-L levels, it is possible that in serum samples this correlation could be finally obtained. I wonder if the authors have considered this possibility. The inclusion of blood samples in this study could reinforce the findings obtained and they would provide a more defined role of CXCL12 in the progression of the disease, even if serial samples are obtained and analyzed.

In addition and probably due to the potential involvement of CXCL12 in the neuroinflammatory response in the disease, a significant increase in CXCL12 levels is observed in the CSF of MS and sALS patients. This point needs further discussion in the manuscript.

Finally, the authors could explain why they have selected only sporadic cases and not familial cases. Some mutations related to the disease are also present in sporadic cases.

Author Response

We have not analyzed in the present study the expression levels of CXCL12 in blood because the samples were not available.  

We have discussed the benefits and limitations of increased levels of CXCL12 in sALS and MS. 

We have selected only sporadic cases to not introduce possible bias linked to expansions and mutations.  

We thank the reviewers for their comments and suggestions

Reviewer 2

Main issues:

  1. How was the diagnosis of ALS made- what criteria were used?

El Escorial criteria updated. Now cited and referenced in the new version

  1. bvFTD has a clear overlap with ALS so it is concerning no association was found. In fact in Fig 6 it shows FTD does not overlap at all. How was bvFTD defined?  

bvFTD was defined following the paper of Raskowski et al., now cited in the new version.

sALS cases did not suffered from dementia

  1. The average specificity with overlap with controls and association with MS, and the lack of association with disease progression/ALSFRS suggests there are limitations in the utility of claiming CXCL12 as a disease biomarker

We agree in that CXCRL12 levels in the CSF are not specific of sALS as they are also increased in the CSF of MS patients. We have softened the previous concept of specific biomarker of disease, and have discussed the benefits and limitations of this marker in the context of the clinical data in every patient.  

  1. The reason for choosing CXCL12 in the spinal cord tissue and not looking at the other proteins could deserve more explanation as the AUC are similar.   

As stated in the text we have chosen this particular marker at present, but we are planning to assess other putative markers in the future. In any case, the list of possible candidates is available to the reader, and this information can be used by other researchers in the field.   

  1. YKL40 in particular and even NF-L are not clearly established as biomarkers and largely remain at a research level so all references to these should be softened and perhaps just restrict commentary to the novel proteins.

This has been softened in the new version, and deleted in the abstract  

  1. I'm assuming the two ALS cohorts are separate ie the 15 aren't included in the larger sample?

Exactly. This point is clarified in material and methods and results.  

  1. The CSF collection section describes the larger cohort and I'm uncertain who the initial 15 ALS subjects/controls were? "

The initial 15 cases are part of the first cohort. This is clarifies in the new version.

  1. Early disease stage" is claimed with "CSF collection about 7-12 months" from onset of symptoms - should give mean and range. It is unclear which of the two cohorts this applies to.  

Mean and range are now included

  1. Detail of disease progression needs to be spelt out- eg was the ALSFRS used (as it has utility for measuring progression); F/U detail including survival could be given; How long were subjects followed for; I'm uncertain what "clinical evolution" means.

ALSFRS was used to monitor disease progression in every case

The survival times and the time the patients were followed are included.

“Clinical evolution” has been deleted

  1. WRT defining spinal onset - in Line 313 - presume they mean scoring 3/4 in the corresponding limb subscales?  

Yes, this is clarified in the text

  1. Minor:   In the first cohort, female/male of 29/14 is very unusual for ALS where typically slightly more males than females appear in most cohorts? Were any of the ALS subjects positive for a genetic diagnosis?

This is a curious feature; we have mentioned this point in the text. However, the results were similar in the first and second cohort.

ALS cases did not carry C9orf72 expansions, TARDP and SOD1 mutations. This information is added in the text. 

  1. A few minor typos eg line 175 "no" 704 "de" Table 2 - Debut is an odd way to describe site of onset.

Corrected

Reviewer 3 Report

Main issues: How was the diagnosis of ALS made- what criteria were used? bvFTD has a clear overlap with ALS so it is concerning no association was found. In fact in Fig 6 it shows FTD does not overlap at all. How was bvFTD defined?   The average specificity with overlap with controls and association with MS ,and the lack of association with disease progression/ALSFRS suggests there are limitations in the utility of claiming CXCL12 as a disease biomarker   The reason for choosing CXCL12 in the spinal cord tissue and not looking at the other proteins could deserve more explanation as the AUC are similar.    YKL40 in particular and even NF-L are not clearly established as biomarkers and largely remain at a research level so all references to these should be softened and perhaps just restrict commentary to the novel proteins   I'm assuming the two ALS cohorts are separate ie the 15 aren't included in the larger sample?  The CSF collection section describes the larger cohort and I'm uncertain who the initial 15 ALS subjects/controls were? " Early disease stage" is claimed with "CSF collection about 7-12 months" from onset of symptoms - should give mean and range. It is unclear which of the two cohorts this applies to.   Detail of disease progression needs to be spelt out- eg was the ALSFRS used (as it has utlility for measuring progression); F/U detail including survival could be given; How long were subjects followed for; I'm uncertain what "clinical evolution" means. WRT defining spinal onset - in Line 313 - presume they mean scoring 3/4 in the corresponding limb subscales?   Minor:   In the first cohort, female/male of 29/14 is very unusual for ALS where typically slightly more males than females appear in most cohorts? Were any of the ALS subjects positive for a genetic diagnosis?   A few minor typos eg line 175 "no" 704 "de" Table 2 - Debut is an odd way to describe site of onset.

Author Response

We thank the reviewers for their comments and suggestionsReviewer 2Main issues: 1. How was the diagnosis of ALS made-what criteria were used? El Escorial criteria updated. Now cited and referenced in the new version2. bvFTDhas a clear overlap with ALS so it is concerning no association was found. In fact in Fig 6 it shows FTD does not overlap at all. How was bvFTD defined? bvFTD was defined following the paper of Raskowski et al., now cited in the new version. sALS cases did not suffered from dementia3. The average specificity with overlap with controls and association with MS,and the lack of association with disease progression/ALSFRS suggests there are limitations in the utility of claiming CXCL12 as a disease biomarker We agree in that CXCRL12 levels in the CSF are not specific of sALS as they are also increased in the CSF of MS patients. We have softened the previous concept of specific biomarker of disease, and have discussed the benefits and limitations of this marker in the context of the clinical data in every patient. 4. The reason for choosing CXCL12 in the spinal cord tissue and not looking at the other proteins could deserve more explanation as the AUC are similar.As stated in the text we have chosen this particular marker at present, but we are planning to assess other putative markers in the future. In any case, the list of possible candidates is available to the reader, and this information can be used by other researchers in the field. 5. YKL40 in particular and even NF-L are not clearly established as biomarkers and largely remain at a research level so all references to these should be softened and perhaps just restrict commentary to the novel proteins. This has been softened in the new version, and deleted in the abstract6. I'm assuming the two ALS cohorts are separate ie the 15 aren't included in the larger sample?Exactly. This point is clarified in material and methods and results. 7. The CSF collection section describes the larger cohort and I'm uncertain who the initial 15 ALS subjects/controls were?" The initial 15 cases are part of the first cohort. This is clarifies in the new version.8. Early disease stage" is claimed with "CSF collection about 7-12 months" from onset of symptoms -should give mean and range. It is unclear which of the two cohorts this applies to. Mean and range are now included9. Detail of disease progression needs to be spelt out-eg was the ALSFRS used (as it has utility for measuring progression); F/U detail including survival could be given; How long were subjects followed for; I'm uncertain what "clinical evolution" means. ALSFRS was used to monitor disease progression in every caseThe survival times and the time the patients were followed are included. Clinical evolutionhas been deleted10. WRT defining spinal onset -in Line 313 -presume they mean scoring 3/4 in the corresponding limb subscales? Yes, this is clarified in thetext

11. Minor: In the first cohort, female/male of 29/14 is very unusual for ALS where typically slightly more males than females appear in most cohorts? Were any of the ALS subjects positive for a genetic diagnosis? This is a curious feature; we have mentioned this point in the text. However, the results were similar in the first and second cohort.ALS cases did not carry C9orf72 expansions, TARDP and SOD1 mutations. This information is added in the text.12. A few minor typos eg line 175 "no" 704 "de" Table 2 -Debut is an odd way to describe site of onset.Corrected

Round 2

Reviewer 2 Report

The authors have explained all the suggested comments. I understand the lack of inclusion of blood samples but they could be helpful in future studies as well as the inclusion of genetic cases.

Reviewer 3 Report

nil